# Post-decision biases reveal a self-consistency principle in perceptual inference

**Long Luu[1], Alan A Stocker[1,2]***

[1]Department of Psychology, University of Pennsylvania, Philadelphia, United States; [2]Department of Electrical and Systems Engineering, University of Pennsylvania, Philadelphia, United States

**Abstract** Making a categorical judgment can systematically bias our subsequent perception of the world. We show that these biases are well explained by a *self-consistent Bayesian observer* whose perceptual inference process is causally conditioned on the preceding choice. We quantitatively validated the model and its key assumptions with a targeted set of three psychophysical experiments, focusing on a task sequence where subjects first had to make a categorical orientation judgment before estimating the actual orientation of a visual stimulus. Subjects exhibited a high degree of consistency between categorical judgment and estimate, which is difficult to reconcile with alternative models in the face of late, memory related noise. The observed bias patterns resemble the well-known changes in subjective preferences associated with cognitive dissonance, which suggests that the brain's inference processes may be governed by a universal self-consistency constraint that avoids entertaining 'dissonant' interpretations of the evidence.

DOI: https://doi.org/10.7554/eLife.33334.001

**\*For correspondence:**
astocker@psych.upenn.edu

**Competing interests:** The authors declare that no competing interests exist.

## Introduction

We make thousands of decisions every day based on sensory information - where to walk, who to greet, and what to eat. How such perceptual decisions are formed through the integration and evaluation of sensory evidence has been extensively studied at the behavioral, computational, and neural level. Helmholtz has described the perceptual decision process as a form of statistical inference (*Helmholtz, 1867*; *Westheimer, 2008*), referring to the fact that sensory information is inherently ambiguous and noisy (*Kersten et al., 2004*; *Rust and Stocker, 2010*). This idea is supported by the recent success of Bayesian statistics in formulating accurate models of human judgments in perception (e.g., *Curry, (1972)*; *Knill and Richards, (1996)*; *Geisler and Diehl, (2002)*; *Körding and Wolpert, (2004)*; *Stocker and Simoncelli, (2006)*; *Wei and Stocker, (2015)*). One common conclusion of this large body of work is that humans typically act like rational decision-makers that trade prior expectations about the world against uncertainty in the sensory evidence in order to optimize perceptual accuracy. However, most of these previous studies considered perceptual judgments as independent, isolated events, which starkly contrasts with real-world situations where perceptual judgments are often embedded in a sequence of other judgments involving the same sensory evidence. Indeed, it is well-known in social psychology and economics that decisions are not independent and can impact subsequent cognitive judgments, leading to seemingly irrational behavior (*Tversky and Kahneman, 1974*; *Kahneman and Tversky, 1979*; *Ariely, 2008*). For example, by making a choice between two equally valued alternatives, subjects tend to adjust their ratings of the two alternatives such that the new values become more consistent with their decision (*Brehm, 1956*; *Sharot et al., 2010*; *Egan et al., 2010*) (but see *Chen and Risen, (2010)*; *Izuma and Murayama,*

*(2013)*). This has been described as a strategy with which humans resolve a state of so-called 'cognitive dissonance' (*Festinger, 1957*; *Festinger and Carlsmith, 1959*). However, little is known about whether decisions also interact with the interpretation of sensory information in subsequent perceptual judgments. A recent study found that a perceptual decision can bias the evidence accumulation process by reducing a subject's sensitivity to subsequent sensory information (*Bronfman et al., 2015*). Similarly, data from two other studies may be interpreted in a way that suggests that perceptual decisions can bias subjects' percept of a stimulus variable such that the percepts become more consistent with their preceding decisions (*Stocker and Simoncelli, 2007*) – although this interpretation differs from that of the original authors (*Baldassi et al., 2006*; *Jazayeri and Movshon, 2007*). Overall, while previous experimental results suggest that categorical decisions interact with subsequent perceptual judgments, the findings have been diverse, and their interpretation rather loose. Most of all, we are lacking a clear quantitative model explanation that connects the various experimental findings.

Here, we tested the hypothesis that an observer's inclination to maintain a self-consistent, hierarchical interpretation of the world leads to the observed post-decision biases in perceptual judgments. We expressed this hypothesis with a *self-consistent Bayesian observer model*, which assumes that a subject's estimate is not only conditioned on the sensory evidence but also on the subject's preceding categorical judgment. The model extends our previous formulation (*Stocker and Simoncelli, 2007*) as it takes into account that sensory information in working memory degrades over time, which has important implications with regard to the behavioral benefits of the proposed hypothesis. We validated the model with three different psychophysical experiments that were based on a perceptual task sequence in which subjects were asked to perform a categorical orientation judgment followed by an orientation estimate of a visual stimulus. Results from the first experiment, using a similar design as the original experiment by *Jazayeri and Movshon, (2007)*, suggest that post-decision biases are general to different low-level visual stimuli. The other two experiments were targeted to specifically test key assumptions of the model such as the probabilistic dependence of the biases on sensory priors. We show that the self-consistent observer model accurately accounts not only for the data from our experiments but also from a previous study (*Zamboni et al., 2016*). The similarity to well documented bias effects in social psychology and economics suggests that the proposed self-consistent model may provide a general framework for understanding sequential decision-making.

## Results

The first goal was to obtain a quantitative characterization of how the perceptual inference process is affected by a preceding, categorical judgment (*Figure 1a*). In Experiment 1 subjects first indicated whether the overall orientation of an array of lines was clockwise (cw) or counter-clockwise (ccw) relative to a discrimination boundary (discrimination task), and then had to reproduce their perceived overall stimulus orientation by adjusting a reference line (estimation task). The experimental design was similar to that of a previous experiment (*Jazayeri and Movshon, 2007*) (see also *Zamboni et al., (2016)*) with the notable exceptions that we used an orientation rather than a motion stimulus, and that subjects had to perform the estimation task in every trial rather than only in one third of the trials (*Figure 1b*). We found that subjects' perceptual behavior was very similar to the findings of these previous studies. Discrimination performance monotonically depended on stimulus noise. Furthermore, reported stimulus orientations showed clear repulsive biases away from the discrimination boundary with larger biases for larger stimulus noise and orientations closer to the boundary (*Figure 1c*). This behavior is fully revealed in the overall distributions of subjects' estimates for every noise level (*Figure 1d*; see *Figure 1—figure supplement 1* for individual subjects): The distributions exhibit a characteristic bimodal shape for stimulus orientations close to the decision boundary, with subjects' estimates biased away from the decision boundary toward the side that corresponds to their preceding discrimination judgment. Together with previous findings (*Jazayeri and Movshon, 2007*; *Zamboni et al., 2016*), the results of Experiment 1 suggest that the observed post-decision biases are independent of the specific type of stimulus used. Also, they indicate that subjects' anticipation of the frequency of the estimation task (every trial *vs.* only in 1/3 of the trials) does not play a role in causing the biases.

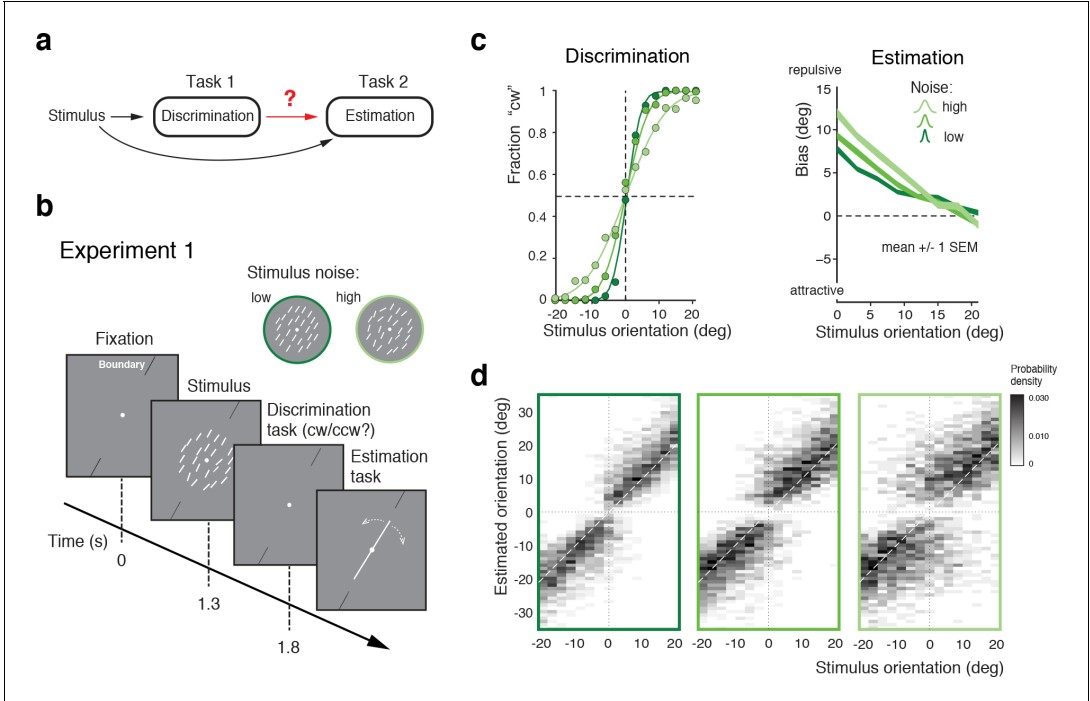

**Figure 1.** Post-decision biases in a perceptual task sequence. (**a**) Perceptual decision-making in a discrimination-estimation task sequence: Does a discrimination judgment causally affect a subject's subsequent perceptual estimate? (**b**) Experiment 1: After being presented with an orientation stimulus (array of lines), subjects first decided whether the overall array orientation was clockwise (cw) or counter-clockwise (ccw) of a discrimination boundary, and then had to estimate the actual orientation by adjusting a reference line with a joystick. Different stimulus noise levels were established by changing the orientation variance in the array stimulus. (**c**) Psychometric functions and estimation biases (combined subject). Estimation biases are only shown for correct trials and are combined across cw and ccw directions. Subjects show larger repulsive biases the noisier the stimulus and the closer the stimulus orientation was to the boundary. (**d**) Distributions of estimates for the three stimulus noise levels tested, plotted as a function of stimulus orientation relative to the discrimination boundary (combined subject). Estimates are clearly biased away from the discrimination boundary forming a characteristic bimodal pattern.

DOI: https://doi.org/10.7554/eLife.33334.002

The following figure supplement is available for figure 1:

**Figure supplement 1.** Full distributions of individual subjects' estimates in Experiment 1.
DOI: https://doi.org/10.7554/eLife.33334.003

## The self-consistent observer model

How are these post-decision biases explained? A Bayesian observer that regards the task sequence as two independent inference processes does not predict the biases. The observer uses Bayesian statistics to determine the correct categorical judgment (e.g., 'cw') based on the stimulus response of a population of sensory neuron, and does the same to infer the best possible estimate of the stimulus orientation (*Figure 2a*). Consequently, this observer's discrimination judgment does not affect the estimation process; orientation estimates are unimodally distributed around the true stimulus orientation and do not exhibit the characteristic bimodal pattern that we have observed in Experiment 1 (*Figure 1d*). In the context of this paper, we refer to this model as the 'independent' observer.

In contrast, we propose an observer model that regards the two tasks as causally dependent (*Figure 2b*); that is, by making the discrimination judgment the observer constrains its subsequent estimation process to consider only those stimulus orientations that are consistent with the judgment. It is as if the observer regards its own, subjective discrimination judgment as an objective fact. Such behavior seems irrational (obviously, the judgment could be incorrect) and furthermore leads to characteristic estimation biases away from the discrimination boundary. It has the advantage, however, that the observer's perceptual inference process remains self-consistent throughout the entire task sequence at any moment in time. We refer to this model as the 'self-consistent' observer. It can be formulated as a conditioned Bayesian model (*Stocker and Simoncelli, 2007*) that jointly

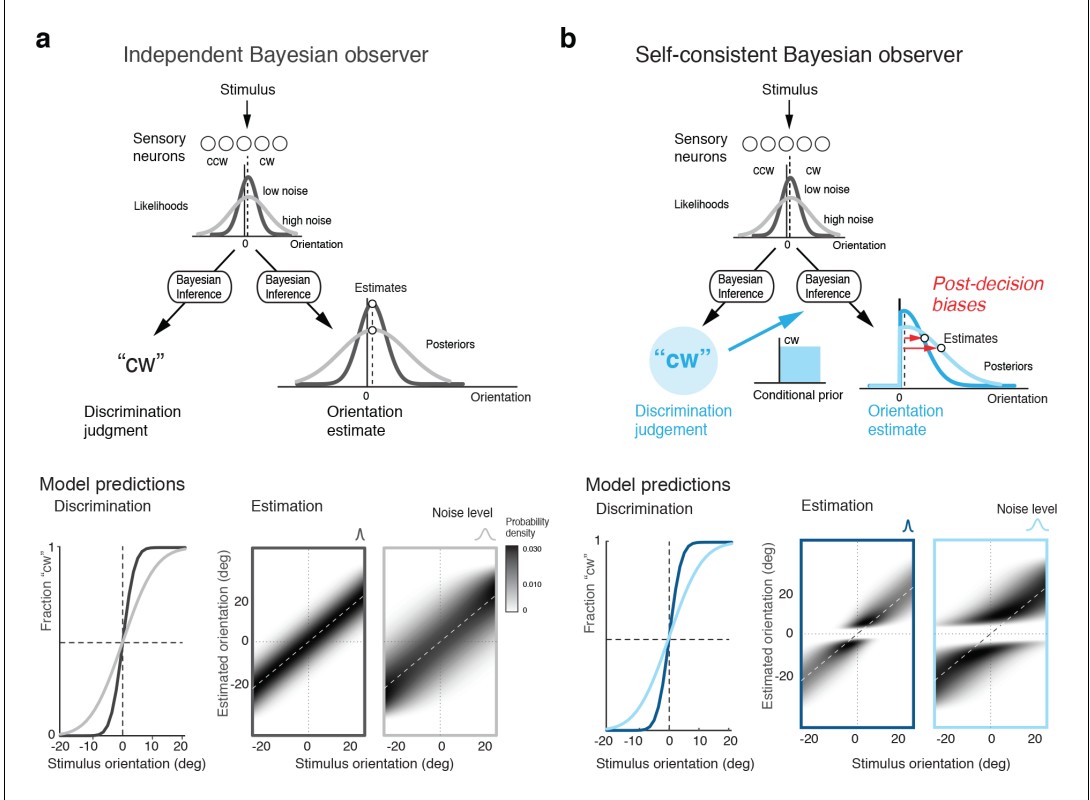

**Figure 2.** Bayesian observer models for the perceptual task sequence. (**a**) The discrimination judgment does not affect the estimated stimulus orientation for an observer who considers both tasks independently. (**b**) In contrast, the self-consistent observer imposes a causal dependency such that the judgment in the discrimination task (e.g., 'cw') conditions the estimation process in form of a choice-dependent prior. It effectively sets the posterior probability to zero for any orientation value that is inconsistent with the preceding discrimination judgment. The truncated posterior distribution, together with a loss function that penalizes larger estimation errors stronger than smaller ones, leads to the characteristic bimodal distribution pattern. Note, however, that this basic formulation is not quite sufficient to explain some details of the estimation data (**Figure 1d**).
DOI: https://doi.org/10.7554/eLife.33334.004

accounts for subjects' behavior in both the discrimination and the estimation task. However, a closer comparison between the predicted (**Figure 2b**) and the measured distribution of the estimates (**Figure 1d**) reveals that this basic formulation does not capture all details of the data.

We formulated the self-consistent observer model as a two-step inference process over the extended hierarchical generative model shown in **Figure 3a**: Based on a noisy sensory signal $m$, the observer first infers the category $C \in \{'cw', 'ccw'\}$ by performing the discrimination task and then infers the stimulus orientation $\theta$ in the estimation task. Because the stimulus has long disappeared by the time the observer performs the estimation task, we assume that estimation of $\theta$ must rely on a noisy memory recall $m_m$ of the sensory signal $m$. Inference on $\theta$ is then conditioned on the preceding discrimination judgment (e.g., $\hat{C} = 'cw'$), which results in the characteristic repulsive estimation biases. Finally, we also took into account that subjects' report of their perceived stimulus orientation is corrupted by motor noise. We measured motor noise for every subject in a control experiment (see **Figure 3—figure supplement 1**) and subsequently used these measured values for all model fits and comparisons. The self-consistent observer model provides a full account of both the observer's discrimination judgment and orientation estimate in each trial, and is thus jointly predicting a subject's psychometric function as well as the distribution of their orientation estimates.

**Figure 3b** shows the model fit to the data from Experiment 1 for the combined subject. The stimulus noise level determines both the slope of the psychometric curves in the discrimination task and, in combination with the memory noise level, the magnitude of the bias in the estimation task, which is well predicted by the model. A comparison between the distributions of subjects' estimates and model estimates fully reveals the extent to which the model accurately accounts for the observed

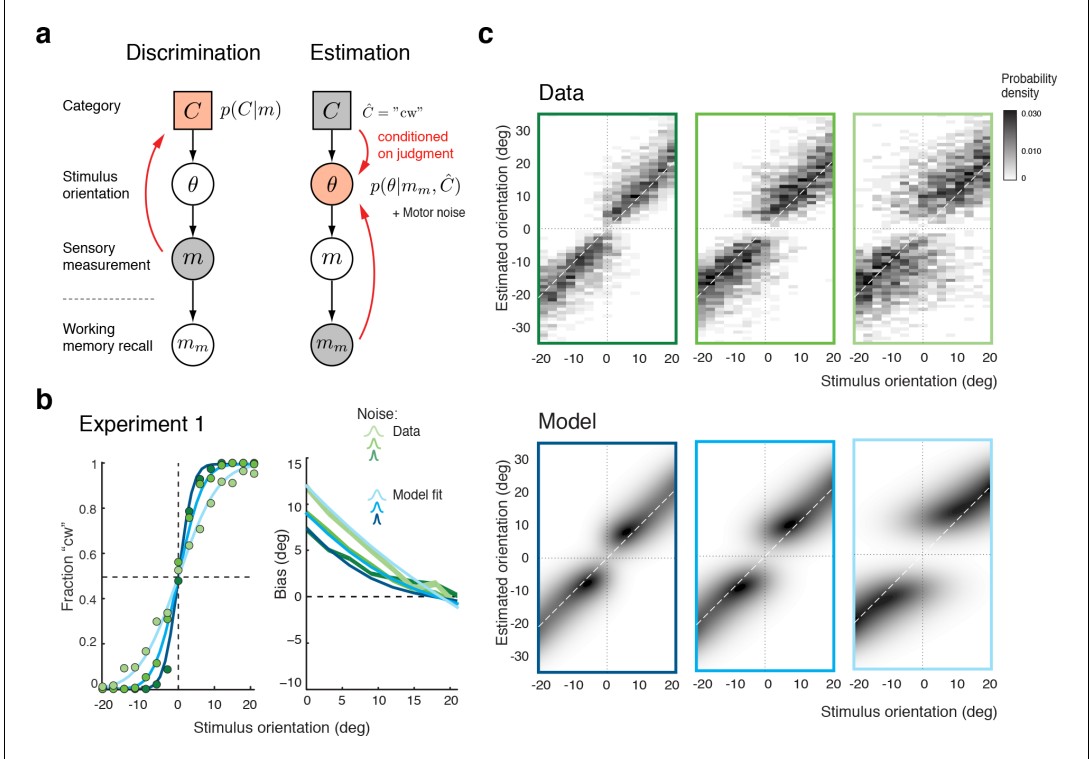

**Figure 3.** The self-consistent Bayesian observer model. (**a**) Directed graph representing the generative hierarchical model: Sensory measurement $m$ is a noisy sample of stimulus orientation $\theta$. Every $\theta$ belongs to one of two categories $C \in \{'cw', 'ccw'\}$. Given an observed $m$, the self-consistent model first performs inference over $C$ (discrimination task), and then infers the value of $\theta$ *conditioned* on the preceding discrimination judgment (e.g., $\hat{C} = 'cw'$) (estimation task). Inference for the estimation task is assumed to be based on a noisy memory recall $m_m$ of the sensory measurement $m$. Conditioning on the categorical choice sets the posterior $p(\theta | m_m, \hat{C})$ to zero for all values of $\theta$ that do not agree with the choice. This shifts the posterior probability mass away from the discrimination boundary and results in the repulsive post-decision biases for any loss function that more strongly penalizes large errors than small ones. Because subjects were instructed to provide estimates as accurate as possible we assumed a loss function that minimizes mean squared-error ($L_2$ loss). (**b**) We jointly fit the observer model to all discrimination-estimation data pairs of the combined data across all subjects in Experiment 1 (combined subject). (**c**) The model not only predicts the mean estimation bias (as shown in (**b**)) but also the entire distributions of estimates, including those trials where discrimination judgments were incorrect. Data and model show the characteristic bimodal pattern for orientation estimates. Each column corresponds to one of the three stimulus noise conditions.

DOI: https://doi.org/10.7554/eLife.33334.005

The following figure supplements are available for figure 3:

**Figure supplement 1.** *Measured motor noise of individual subjects.*
DOI: https://doi.org/10.7554/eLife.33334.006

**Figure supplement 2.** Histogram plots of the orientation estimates together with the model fit for Experiment 1 (combined subject).
DOI: https://doi.org/10.7554/eLife.33334.007

human behavior (*Figure 3c*; See *Figure 3—figure supplement 2* for a histogram representation). Note that all model predictions in this paper are the result of a joint fit to both the measured psychometric functions of the discrimination task and the estimation distribution.

The self-consistent observer model can also account for the substantial individual differences in behavior across subjects. While individual bias patterns are all repulsive, they vary across subjects both in shape and magnitude (*Figure 4a*). These variations are well captured by the model and reflected in individual differences in the fit parameter values such as the prior width and the level of sensory noise (*Figure 4b*; see *Figure 4—figure supplement 1* for a goodness-of-fit analysis). Interestingly, all subjects seemed to substantially overestimate the width of the stimulus prior as compared to the true stimulus distribution. This did not come entirely as a surprise because subjects were never explicitly informed about the stimulus range and thus had to learn it over the course of

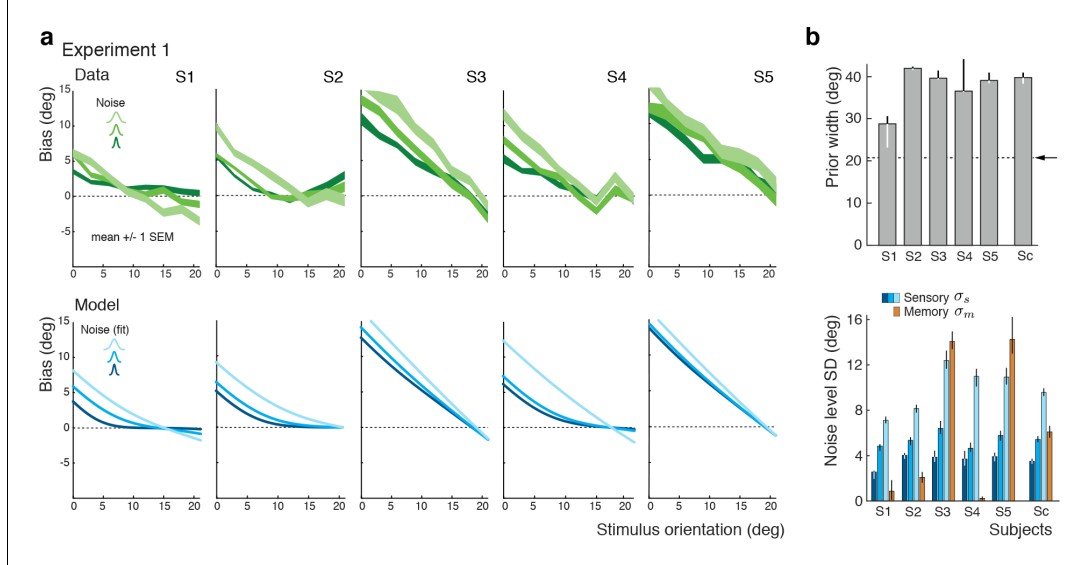

**Figure 4.** Experiment 1: Data and model fits for individual subjects. (**a**) Individual subjects (S1 non-naïve) showed substantial variations in their bias patterns (green curves). These variations are well explained by individual differences in the fit parameter values of the self-consistent model (blue curves). For example, the width of the prior directly determines the location where the bias curves intersect with the x-axis. (**b**) Fit prior widths $w_p$ and noise levels for the five individual subjects plus the combined subject (Sc). Subjects' prior widths suggest that they consistently overestimated the actual stimulus range in the experiment (± 21 degrees; arrow). For all subjects, fit sensory noise $\sigma_s$ was comparable and monotonically dependent on the actual stimulus noise. Memory noise $\sigma_m$ was mostly small as expected, yet dominated for subjects S3 and S5. These two subjects performed poorly in the estimation task, suggesting that they were not trying to provide an accurate orientation estimate but simply pointed the cursor to roughly the middle of the stimulus range on the side of the discrimination boundary they picked in the discrimination task. The resulting bias curves are basically independent of the stimulus noise and have a slope of approximately −1. The model captured this behavior by assuming that the sensory information was 'washed out' with a large amount of memory noise. The full model also contained a motor noise component that was determined for each subject in a separate control experiment. All errorbars represent the 95% confidence interval computed over 100 bootstrapped sample sets of the data. See Materials and methods for details.

DOI: https://doi.org/10.7554/eLife.33334.008

The following figure supplement is available for figure 4:

**Figure supplement 1.** *Goodness-of-fits for Experiment 1.*

DOI: https://doi.org/10.7554/eLife.33334.009

the experiment. Consistent with this interpretation is the fact that the subject with the most accurate estimate of the prior distribution was the only non-naïve subject S1 who had plenty of extra exposure to the stimulus range through the participation in various pilot experiments. Extracted noise levels differ across subjects with the worst subject being approximately twice as noisy as the best (non-naïve subject S1), yet consistently increase for increasing stimulus noise levels.

## Probing the self-consistent observer model

We ran two additional experiments that were designed to specifically probe two key features of the self-consistent model: Experiment 2 was aimed at testing how subject's orientation estimates were dependent on their precise knowledge of the stimulus prior and thus were consistent with the results of Bayesian inference; Experiment 3 examined whether subjects indeed treated their discrimination judgments as if they were correct. We recruited a new set of subjects (S6–S9, plus S1) that performed both experiments. By jointly fitting the data from both experiments, we also tested how well the model can generalize across tasks (Fits and model parameters for subject S1 are the result of a joint fit to the data from all three experiments).

Experiment 2 was identical to Experiment 1 except that at the beginning of each trial, subjects were explicitly reminded of the total range within which the stimulus orientation would occur in the trial (*Figure 5a*). Our assumption was that an explicit display of the stimulus range provided subjects with a better and presumably narrower representation of the stimulus distribution (given that

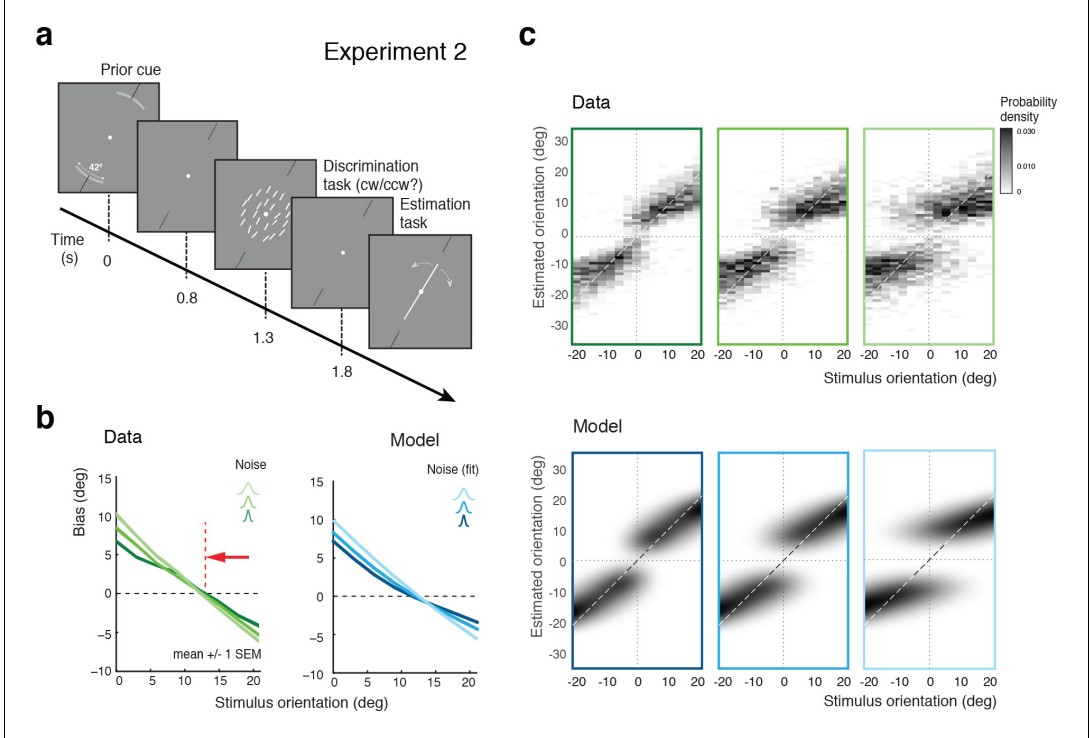

**Figure 5.** Effect of the stimulus prior. (**a**) Experiment 2 was identical to Experiment 1 except that at the beginning of each trial, subjects were shown the total range within which the stimulus orientation would occur in the trial (gray arc, subtending ± 21 degrees). (**b**) We hypothesize that reminding subjects of the exact stimulus range at the beginning of each trial helps them to form a more accurate (and more narrow) representation of their stimulus prior. If subjects' orientation estimates were indeed the result of the conditioned Bayesian inference as assumed by the self-consistent observer model, then the bias curves should shift towards the discrimination boundary. The data support this prediction: Subjects' bias curves (combined subject, see **Figure 7** for individual subjects) are shifted towards the discrimination boundary compared to Exp. 1. (**c**) As with Exp.1, the fit self-consistent model provides an accurate description of the distribution pattern of subjects' orientation estimates.

DOI: https://doi.org/10.7554/eLife.33334.010

The following figure supplement is available for figure 5:

**Figure supplement 1.** Full distributions of individual subjects' estimates in Experiment 2.

DOI: https://doi.org/10.7554/eLife.33334.011

subjects seemed to substantially overestimate the prior in Experiment 1). If so, then the self-consistent observer model would predict a shift of the bias curves' crossover point towards the discrimination boundary. As shown in **Figure 5b**, the measured bias curves indeed show the predicted shift compared to the bias curves measured in Experiment 1 (**Figure 3b**). This shift is also clearly visible in the distributions of the orientation estimates (see **Figure 5—figure supplement 1** for distributions of individual subjects), which is again accurately accounted for by the model (**Figure 5c**).

In Experiment 3, we separated the discrimination judgment from the discrimination task. Subjects were no longer asked to perform the discrimination task but instead were signaled right at the beginning of each trial whether the stimulus orientation would be cw or ccw (**Figure 6a**). Subjects were instructed that this categorical information was always correct, which it was. They then performed an unrelated color discrimination task before finally performing the estimation task. The self-consistent model predicts estimation biases that are basically identical to those of Experiment 2 because it assumes that subjects treat their own judgment as correct when performing the estimation task. Indeed, as shown in **Figure 6b**, subjects' estimation biases are very similar to the biases in Experiment 2 (**Figure 5b**). Because Experiments 2 and 3 were both conducted on the same set of subjects, the results are directly comparable. We can rule out that subjects may have ignored the given category assignment in Experiment 3 and implicitly performed the orientation discrimination task instead. If this were the case, then subjects would have exhibited a large fraction of inconsistent

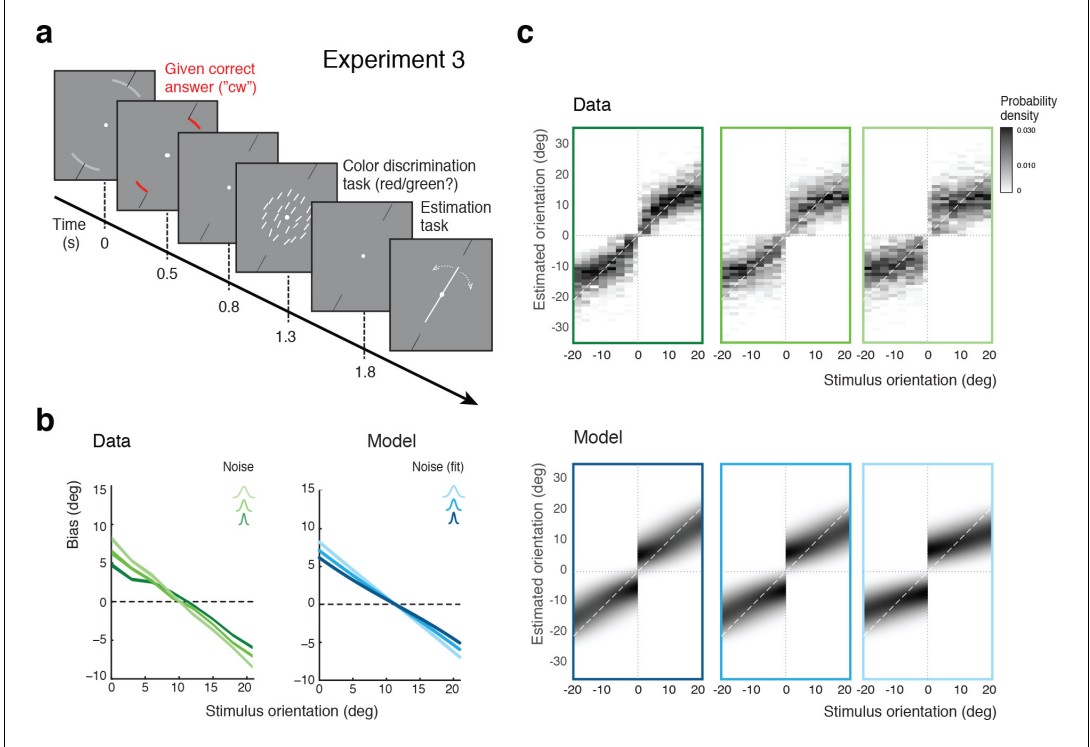

**Figure 6.** Self-made vs. given category assignment. (**a**) Experiment 3: Instead of performing the discrimination judgment themselves, subjects were provided with a cue indicating the correct category assignment right before the stimulus was presented. Then, after stimulus presentation, subjects first performed an unrelated color discrimination task in place of the orientation discrimination task (they needed to remember the randomly assigned color (red/green) of the cue indicating the correct category) before indicating their perceived stimulus orientation. (**b**) According to our model we should see similar estimation biases in Exps. 2 and 3, which is indeed what we found. (**c**) Again, the fit model well accounts for the overall distribution of orientation estimates (combined subject; see *Figure 6—figure supplement 1* for distributions for individual subjects)). Because the discrimination judgment was given and always correct independent of the noise in the sensory measurement *m*, estimates only occurred in the 'correct' quadrants. For the same reason the model also predicts slightly smaller bias magnitudes (compared to Exp.2), which is also matched by the data (see also *Figure 7b*).

DOI: https://doi.org/10.7554/eLife.33334.012

The following figure supplement is available for figure 6:

**Figure supplement 1.** Full distributions of individual subjects' estimates in Experiment 3.

DOI: https://doi.org/10.7554/eLife.33334.013

trials (*i.e.*, trials in which the estimated orientation was not in agreement with the given correct answer) in particular for orientations close to the discrimination boundary. This was not the case as we observed only small fractions of inconsistent trials (4% on average) that were of similar magnitude as the error rates for the (irrelevant) color discrimination task (2%). We discuss these inconsistent trials in more detail in the next sections below.

We again extended our analysis to individual subjects' behavior. *Figure 7a* shows subjects' estimation biases in both experiments as well as the corresponding model predictions based on a joint fit to data from both Experiments 2 and 3. Bias patterns, while quite variable across subjects, are consistent across the two experiments for each subject. This confirms that the impact of the categorical discrimination judgment on the perceived orientation does not depend on whether the judgment was performed by the subjects themselves or not. The model captures both the inter-subject as well as the within-subject variability across the two experiments. Biases are slightly smaller in Experiment 3 compared to Experiment 2 for stimulus orientations close to the boundary. As shown in *Figure 7b*, the model correctly predicts this difference because the self-made discrimination judgments in Experiment 2 are based on the noisy stimulus measurement *m* and therefore can be incorrect, while the category cues in Experiment 3 were always correct. Consequently, the predicted bias curves for Experiment 2 only represent trials for which the sensory measurement *m* was in favor of

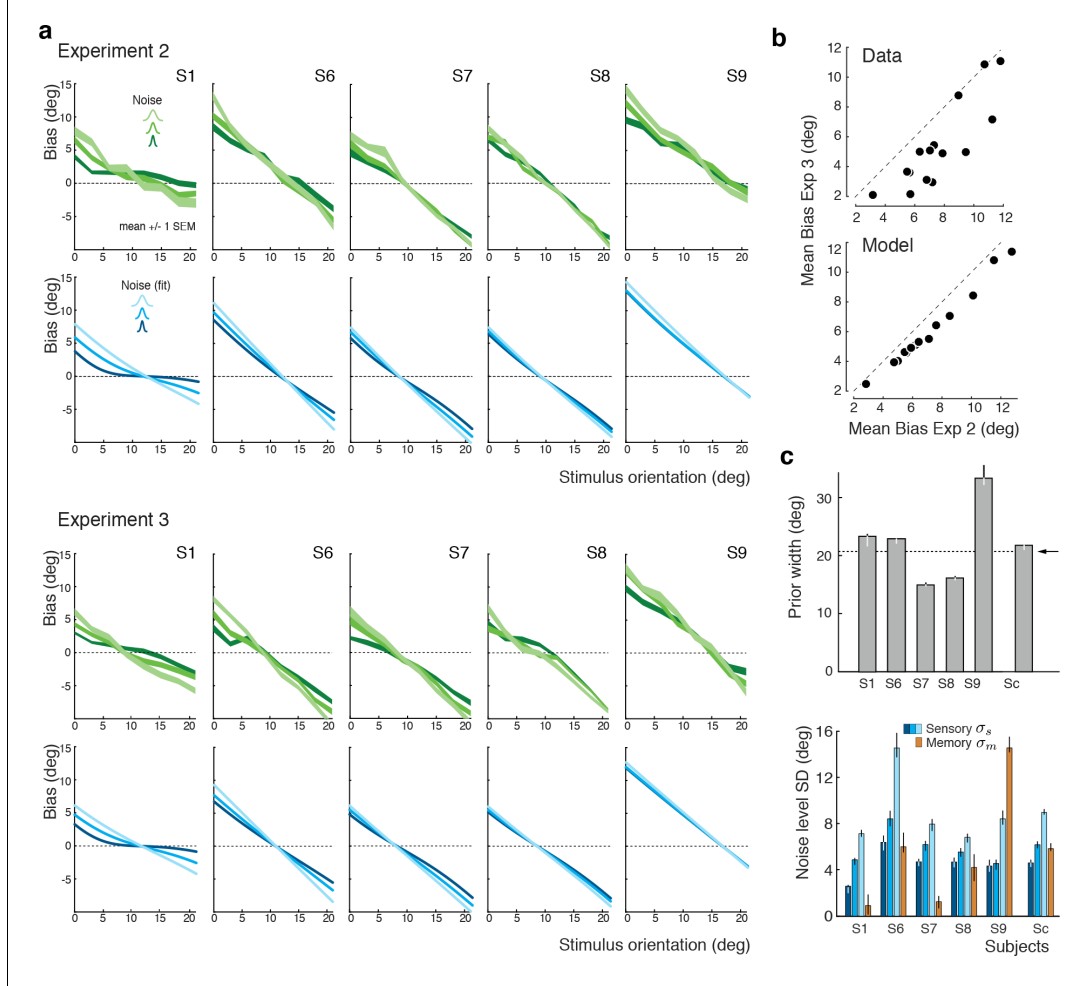

**Figure 7.** Experiments 2 and 3: Joint fit to data for individual subjects. (**a**) Five subjects (S1, S6-9) participated both in Exp. 2 and 3. We performed a joint model fit to the data from both experiments for every subject. Each column shows data (green curves) and fit (blue curves) for a particular subject. As in Exp. 1, the bias pattern across subjects shows substantial variability yet is strikingly similar between the two experiments. (**b**) Comparing the mean biases observed in Exps. 2 and 3 reveals that biases in Exp. 3 are slightly smaller for stimulus orientations close to the boundary. This effect is predicted by the model. (**c**) Fit prior widths $w_p$ and noise levels for individual subjects and the combined subject. Subjects' priors were closer to the experimental distribution than in Exp. 1 because in Exps. 2 and 3 subjects were reminded about the stimulus range at the beginning of each trial. Noise levels were comparable to those in Exp. 1 (for S1 we jointly fit data from all three experiments). Errorbars indicated the 95% confidence interval over 100 bootstrapped samples of the data. See *Figure 7—figure supplement 1* for a goodness-of-fit analysis.

DOI: https://doi.org/10.7554/eLife.33334.014

The following figure supplement is available for figure 7:

**Figure supplement 1.** Goodness-of-fits for Experiments 2 and 3.

DOI: https://doi.org/10.7554/eLife.33334.015

the correct judgment (i.e., $m$ was on the correct side of the boundary) whereas the bias curves for Experiment 3 are computed over all trials. As a result, the biases in Experiment 3 are smaller for stimulus orientations for which there is a substantial chance that the noise pushes the measurements $m$ to the other side of the discrimination boundary. *Figure 7c* shows the fit parameter values for individual subjects. Compared to Experiment 1, the subjective prior widths are substantially smaller and closer to the true stimulus prior width, which suggests that explicitly reminding subjects of the true stimulus distribution in every trial was effective. As in Experiment 1, subjects showed large variations in subjective noise levels although they consistently were monotonic in actual stimulus noise. Fit memory noise levels were relatively small with the notable exception of subject S9 whose poor

performance in the estimation task, quite similar to subjects S3 and S5 in Experiment 1, was picked up by the memory noise parameter.

## Inconsistent trials are due to lapses and motor noise

In a small fraction of trials (on average 4%) subjects' discrimination judgment and their subsequent orientation estimate were not consistent. We can show that these inconsistent trials are not a violation of self-consistent inference but rather can be entirely explained by two common sources of behavioral errors not related to perceptual inference: lapses and motor noise. In fact, we can accurately predict the estimation patterns and individual fractions of inconsistent trials based on model fits to the consistent data, and individual measurements of lapse rates and motor noise.

*Figure 8a* shows the distribution of subjects' estimates (combined data across all subjects and all stimulus noise conditions) in inconsistent trials for all three experiments, together with predictions from the self-consistent observer model for each of the two error sources. Lapses are defined as

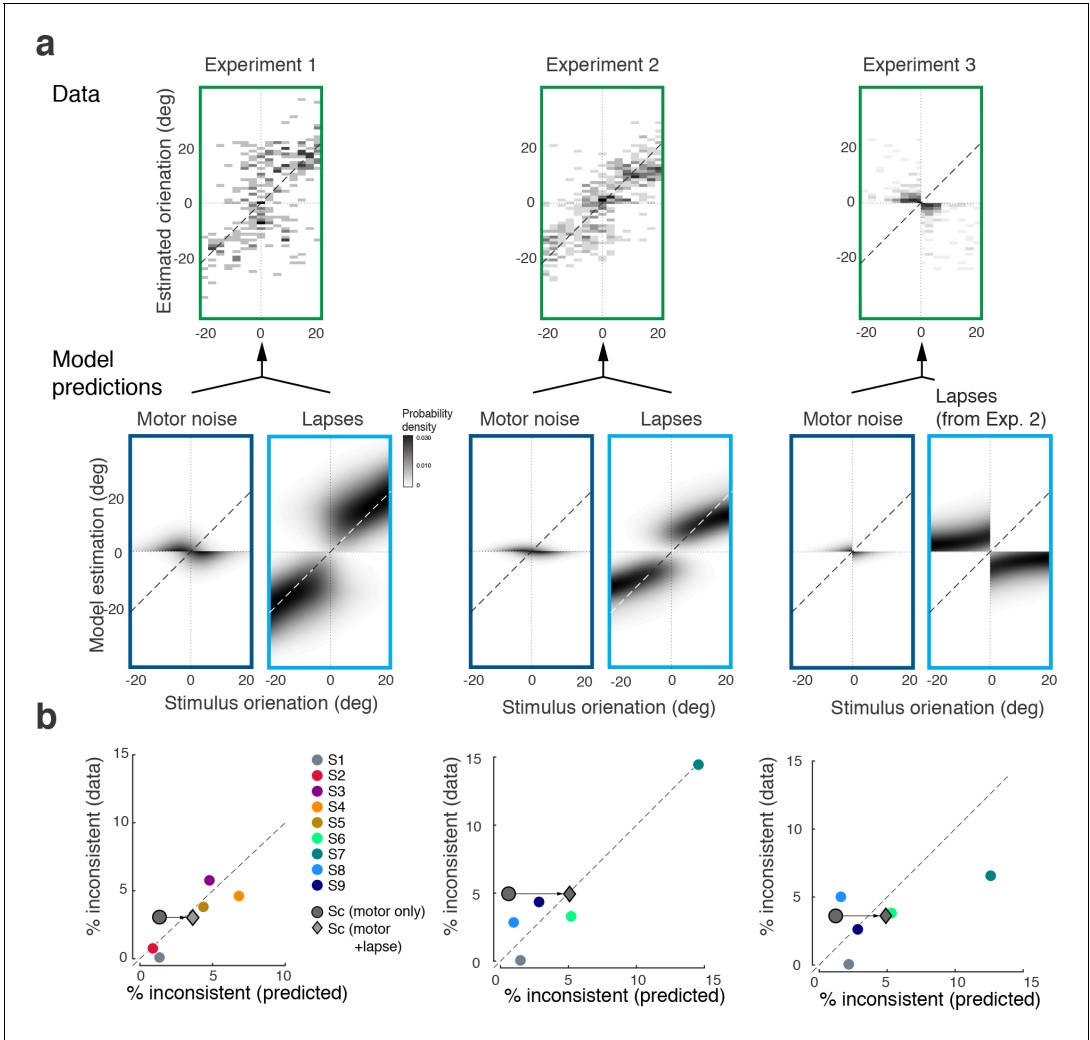

**Figure 8.** Inconsistent trials are due to lapses and motor noise. (a) Distribution of estimates for the small fraction of inconsistent trials (4% of the data) in each experiment (across all subjects and stimulus noise conditions). The estimation patterns can be explained as a weighted superposition of two sources of erroneous, non-perceptual behavior: lapses and motor noise. The self-consistent model well predicts the estimation patterns. All predictions are based on parameter values taken from the model fit to the consistent trial data (see Materials and methods). Lapse rates were extracted from the psychometric functions of the discrimination judgment for the total data. Motor noise was measured in a control experiment (see Materials and methods, *Figure 3—figure supplement 1*). (b) Quantitative predictions for each subject's total fraction of inconsistent trials are compared to the measured fractions. Predictions for the combined subjects suggest that inconsistent trials are mainly due to lapses.
DOI: https://doi.org/10.7554/eLife.33334.016

trials in which subjects by mistake pushed the wrong button in the discrimination judgment (respectively, incorrectly remembered the answer cue in Experiment 3) yet followed up with an orientation estimate that corresponded to their actual judgment. For Experiments 1 and 2, the predicted estimation patterns for lapse trials are similar to the predictions for consistent trials (*Figures 3c* and *5c*), which makes sense since we assumed that subjects performed the estimation task correctly (i.e., were self-consistent) but mistakenly pushed the wrong button in the discrimination task. For Experiment 3, the pattern is different because the misremembered answer cue is always incorrect, and thus subjects' estimates are based on the long tail of the sensory measurement distribution. In contrast, motor noise leads to inconsistent trials when it accidentally deflects subjects' reported orientation estimates to the other side of the discrimination boundary. Thus for all experiments they are predicted to be limited to stimulus orientations close to the boundary. A visual comparison between the measured and the predicted estimation pattern (*Figure 8a*) confirms that the small fraction of inconsistent trials are qualitatively well explained as the combined effect of errors due to lapses and motor noise. Furthermore, we can quantitatively predict individual subjects' overall fraction of inconsistent trials based on their fit model parameter values, and measured lapse rates and motor noise (*Figure 8b*). Analyzing the contribution of each error source further reveals that the majority of inconsistent trials are caused by lapses in the discrimination task.

## Maintaining self-consistency in the face of working memory degradation

To what degree is self-consistent inference a necessary condition for self-consistent behavior? If working memory were perfect (i.e., the sensory signal $m$ and its memory recall $m_m$ are identical) then any reasonable observer model would be self-consistent. However, this is an unlikely scenario because it is fairly well established that continuous visual information in working memory is degrading rather quickly over time (*Wilken and Ma, 2004*; *Bays et al., 2011*). We thus expect working memory degradation to affect perceptual behavior, in particular in Experiments 1 and 2 where the average time between stimulus presentation and the estimation task was on the order of 2–3 s. This is supported by the model fits that revealed non-zero memory noise levels. In order to quantify how challenging working memory noise is for maintaining self-consistency, we computed the fractions of inconsistent trials we would expect without self-consistent inference, based on the fit memory noise levels.

*Figure 9a* shows the predicted fractions of inconsistent trials as a function of stimulus orientation for every subject and stimulus noise condition. The curves reflect the fraction of trials in which sensory measurement $m$ and the working memory recall $m_m$ are on different sides of the reference boundary. Predictions vary for individual subjects yet are typically large in particular for orientations close to the boundary.

A comparison with the observed fractions of inconsistent trials in Experiments 1 and 2 reveals that those are much smaller and relatively independent of the stimulus orientation (combined subject for statistical reasons; *Figure 9b*), in line with our previous conclusion that inconsistent trials predominantly reflect lapses (see *Figure 8*). This is further supported by a comparison with subjects' individual memory noise levels: the predicted fractions are almost perfectly correlated with memory noise whereas no such correlation can be found for the observed fractions (*Figure 9c*). Thus, above analysis suggests that if memory noise is present, the proposed self-consistency constraint is necessary in order to account for the low fractions of inconsistent trials in the data.

## Further validation with existing experimental data

In a recent study, *Zamboni et al., (2016)* run different variations of the original experiment (*Jazayeri and Movshon, 2007*). Specifically, they manipulated the presence and orientation of the discrimination boundary at the time of the estimation task, as well as whether subjects had to explicitly perform the discrimination task or not. We fit our model to this dataset (combined subject) and the results are shown in *Figure 10*. Experiment 1a was an exact replica of the original experiment (*Jazayeri and Movshon, 2007*). The observed bias patterns are consistent with the original results as well as the results from our Experiment 1, and thus well accounted for by our model (*Figure 10a*). Experiment 1b was identical to 1a except that the discrimination boundary was removed right after subjects performed the discrimination task. This manipulation led to an increase in variance and a

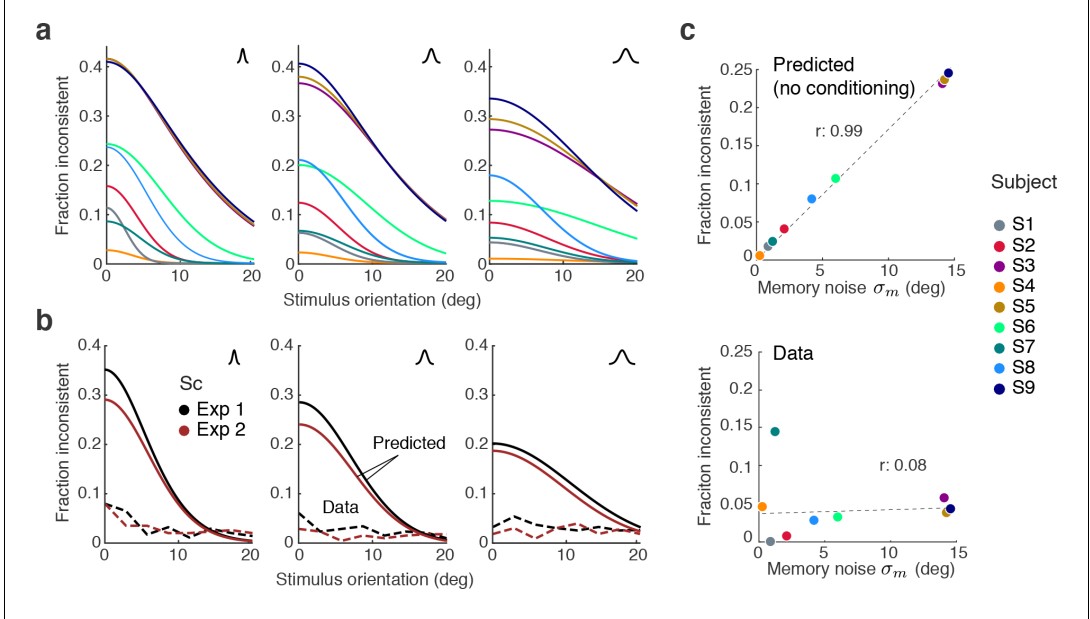

**Figure 9.** Maintaining self-consistency in the face of working memory noise. (**a**) Shown are the predicted fractions of inconsistent trials if orientation estimates are not conditioned on the preceding judgment. These are trials for which the sensory signal $m$ and its memory recall $m_m$ are on different sides of the discrimination boundary. Using the fit model parameters from Exps. 1 and 2, each curve represents the fraction of inconsistent trials as a function of stimulus orientation for every subject (color code on the right). Each panel is for one of the three stimulus noise conditions. These large fractions are predicted for any non-trivial model whose discrimination judgment is based on $m$ and the estimate on $m_m$ but does not condition the estimation process on the preceding discrimination judgment. For simplicity, we did not include lapses and motor error for this analysis, and thus these predictions reflect the direct consistency benefit of conditioning the estimate on the preceding discrimination judgment. (**b**) The actual fractions of inconsistent trials are much lower and relatively independent of stimulus orientation as they are mostly due to lapses (see **Figure 8b**); shown is the combined subject. (**c**) The benefit of self-consistent inference is substantial for larger memory noise; predicted fractions are almost perfectly correlated with the fit memory noise $\sigma_m$ of individual subjects. In comparison, the actual fractions of inconsistent trials are uncorrelated with memory noise levels, in line with our previous analysis showing that they are mainly due to lapses and motor noise.

DOI: https://doi.org/10.7554/eLife.33334.017

loss of bimodality in the distribution of estimates (**Figure 10b**). Interestingly, however, the data are consistently better fit by the self-consistent model than by the independent model that strictly predicts a unimodal distribution, if we assume that the observer had to rely on a noisy memory representation of the discrimination boundary for the estimation task. A detailed inspection of the estimate distributions shows that they are wider the closer the stimulus direction is to the boundary and generally skewed towards the boundary. This suggests that subjects behaved according to the self-consistent observer model, yet the characteristic bimodal estimation pattern is hidden in the extra variance introduced by the uncertainty about the boundary orientation.

In Experiment 2, the boundary orientation was shifted by a small amount (±six degrees) before subjects had to perform the estimation task. Introducing a short blank screen right before the shift ensured that subjects were not aware of this manipulation. In contrast to Experiment 1, subjects were only asked to perform the estimation task. Subjects' estimates still show the same characteristic bimodal distribution although they are shifted according to the boundary shift. This suggests that subjects implicitly performed the discrimination task even though they were not asked to report an explicit judgment, which is supported by the good model fit (**Figure 10c**). Based on these results, we conclude that self-consistent inference takes place at the time of the estimation task, can occur on memorized boundary information, and does not necessarily require an explicit discrimination judgment.

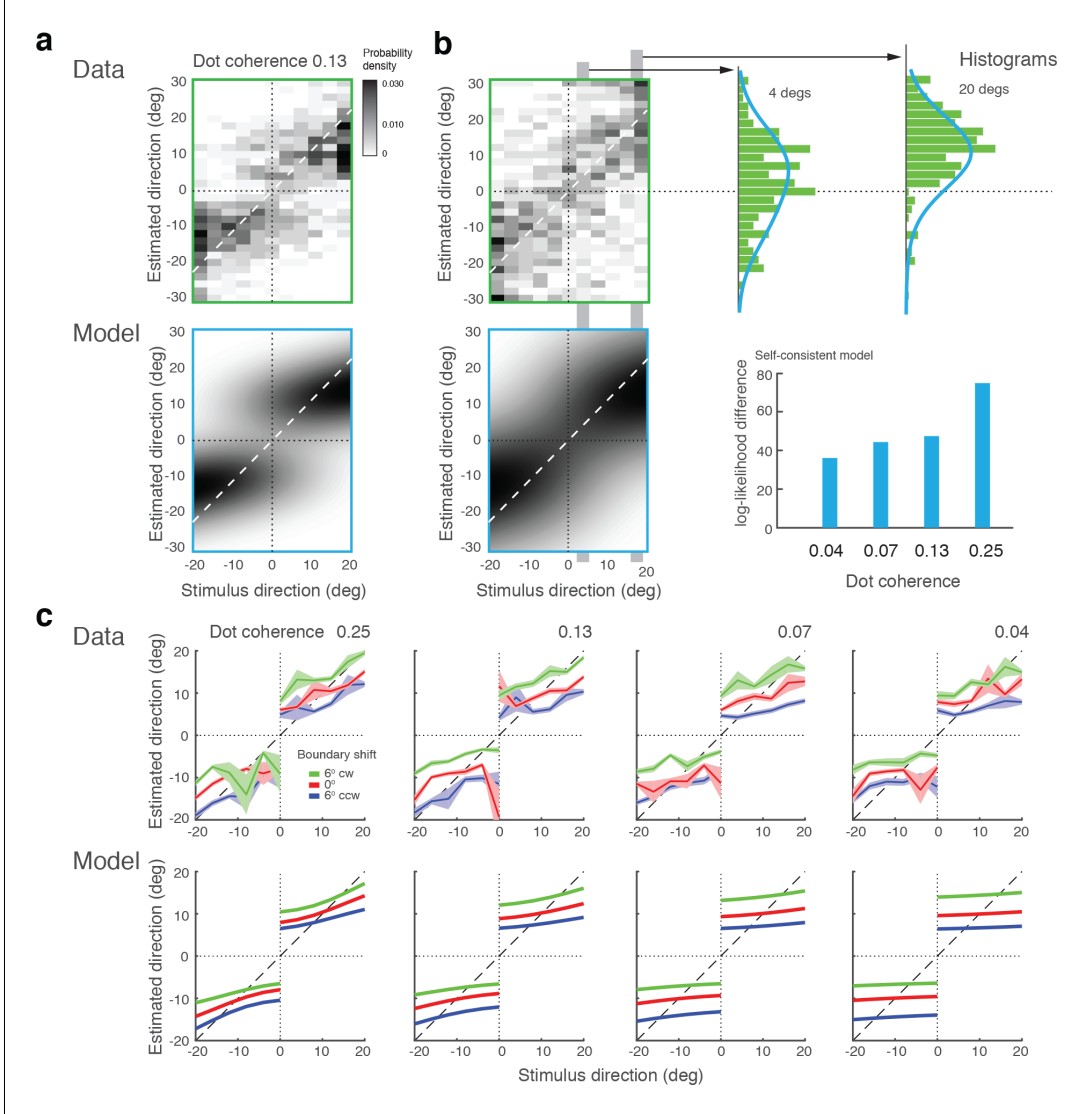

**Figure 10.** Model fits for experimental data by *Zamboni et al., (2016)*. (**a**) Experiment 1a: Exact replication of the original experiment (*Jazayeri and Movshon, 2007*). Exemplarily shown is the estimation data (combined subject) at one stimulus coherence level (0.13) together with our model fit. (**b**) Experiment 1b was identical except that the boundary was not shown during the estimation task. Estimate distributions are no longer bimodal yet the self-consistent observer, relying on a noisy memory of the boundary orientation, consistently better fit the data than the independent observer model (log-likelihood difference). (**c**) Experiment 2 introduced a shift in the boundary orientation right before the estimation task, which subjects were not aware of (±six degrees). Subjects' estimates were shifted accordingly (combined subject). The self-consistent model can account for the shift if we assume that the conditional prior is applied to the shifted boundary orientation. See *Figure 10—figure supplements 1–3* for distributions, fits, and goodness-of-fits for all conditions.

DOI: https://doi.org/10.7554/eLife.33334.018

The following figure supplements are available for figure 10:

**Figure supplement 1.** *Zamboni et al., (2016)* data (Experiment 1, combined subject) and fit with the self-consistent observer model.
DOI: https://doi.org/10.7554/eLife.33334.019

**Figure supplement 2.** Relative log-likelihoods of model fits for *Zamboni et al., (2016)* data.
DOI: https://doi.org/10.7554/eLife.33334.020

**Figure supplement 3.** *Zamboni et al., (2016)* data (Experiment 2, combined subject) and fit with the self-consistent observer model.
DOI: https://doi.org/10.7554/eLife.33334.021

# Discussion

We have shown that in a discrimination-estimation task sequence, the estimated value of a stimulus variable is systematically biased by the preceding discrimination judgment about that variable. We have introduced a self-consistent Bayesian observer model that provides an accurate and complete description of perceptual behavior in such sequential decision-making. The model assumes that the estimates are the result of a Bayesian inference process over a hierarchical generative model, which, however, is conditioned not only on the sensory evidence but also on the subject's judgment in the preceding discrimination task. This guarantees that discrimination judgments and estimates in any given trial are consistent even when the observer must rely on working memory signals that are noisy. We show that an observer that considers the tasks independently will substantially fail to provide the level of consistency observed in the data. With a set of targeted psychophysical experiments we verified that the observed bias pattern generalizes for different low-level visual stimuli (Experiment 1), and validated the self-consistent model by showing that the pattern indeed depends on subjects' knowledge of the stimulus prior (Experiment 2) and that subjects use their own decision as if it was correct (Experiment 3). We further validated the model with existing data from experiments that manipulated the presence and orientation of the discrimination boundary. Successful fits of the proposed observer model to individual subjects data across the various experiments demonstrate the power and accuracy of the model, and its ability to generalize across experimental conditions. Furthermore, the model fits provide a meaningful interpretation of the substantial between-subject differences in behavior in terms of individual differences in noise levels and knowledge of the stimulus prior.

## Alternative interpretations

While various dependencies in sequential perceptual choice tasks have been reported, such as dependencies between decision outcomes (*Fernberger, 1920*; *Senders and Sowards, 1952*; *Gold et al., 2008*; *Fründ et al., 2014*; *Abrahamyan et al., 2016*), decision confidence (*Rahnev et al., 2015*; *Fleming and Daw, 2017*), reaction times (*Laming, 1979*; *Link, 1975*), and error rates (*Vervaeck and Boer, 1980*), the impact of subjects' choices on their immediate subsequent perceptual judgments has not yet been considered a cause for sequential dependencies. Notably, *Jazayeri and Movshon, (2007)* interpreted the reported post-decision biases as the result of a selective read-out strategy by which the brain preferentially weighs signals from those sensory neurons that are most informative with respect to the discrimination task, yet is then compelled to use the same weighted read-out signal when performing the subsequent estimation task. They conclude that a non-uniform read-out profile that more strongly weighs neurons with preferred tuning slightly away from the discrimination boundary could explain the repulsive bias patterns. This is a more mechanistic, neural interpretation, which complicates a direct comparison with our normative computational model. Nevertheless, there is a fundamental difference between this interpretation and our self-consistent model in the way the two perceptual tasks interact: the read-out model proposes a feed-forward process where both tasks are performed independently based on the same weighted sensory signal, whereas our model assumes that feed-back of the categorical judgment is causally affecting the estimation process. Despite this difference, our experimental results alone may not be sufficient to disambiguate between these two interpretations. Experiments 2 and 3 were foremost designed to test the specific aspects of our self-consistent observer model. Although subjects' behavior in these experiments is not compatible with the originally proposed rationale for the particular shape of the read-out profile (optimized for the discrimination task [*Jazayeri and Movshon, 2007*]), relaxing this assumption by also allowing stimulus prior information to determine the shape of the profile may lead to an alternative explanation of our data. Future work must show whether this is true or not. It seems important that such work can establish a principled description of how stimulus prior information ought to be reflected in the weighting function, otherwise the model's explanatory power will be reduced to that of a curve fit. It seems also important that any potential model comparison takes full advantage of the richness of the behavioral data, that is, models should be evaluated based on their ability to account for the entire distribution of subjects' discrimination-estimation response pairs and not only on summary statistics such as bias (*Jazayeri and Movshon, 2007*; *Zamboni et al., 2016*).

More challenging to reconcile with the read-out model, or any other model that does not impose some form of self-consistency constraint, are the experimental results by *Zamboni et al., (2016)* in combination with our consistency analysis (*Figure 9*). The results by *Zamboni et al., (2016)* suggest that a different (or at least adjusted) read-out profile must be applied at the time of the estimation task, which implies that the sensory signal up to that point needed to be stored in some form of working memory. With working memory quickly deteriorating over time (*Wilken and Ma, 2004*; *Bays et al., 2011*), our consistency analysis shows that the observed degree of trial consistency cannot be achieved by an observer model that does not condition the estimate on the discrimination judgment (*Figure 9*). Future research is necessary to validate the levels of working memory noise we have determined with our model.

## General implications for computational and neural models of decision-making

Our results and in particular the proposed self-consistent inference model have broad implications for understanding human decision-making in general. For example, subjects did not distinguish between a decision outcome they generated (as in Experiment 2) and a decision outcome that was given to them (Experiment 3), which implies that for the purpose of performing the estimation task they treat their own subjective judgment as if it was correct. Computationally, this is interesting because on one hand it apparently seems to violate optimal behavior in terms of overall perceptual accuracy (obviously, a subjective judgment can be wrong). On the other hand, however, it guarantees that the observer remains self-consistent throughout the task sequence even when noise is corrupting the sensory information in working memory (*Figure 9*). This is consistent with previous results showing that selectively discarding evidence (a seemingly irrational behavior) can improve performance when decision formation is corrupted by internal neural noise (*Tsetsos et al., 2016*), and thus may be rational after all. Future work is needed to investigate in more detail the impact of self-consistent inference on choice performance and perceptual accuracy.

Furthermore, self-consistent inference can also save computational costs in situations where the task-sequence is more complex. Starting from the top, it reduces a decision tree at every level of the hierarchy by considering only the chosen branch, which can substantially reduce the overall computational complexity and cost associated with solving the inference problem. Self-consistent inference may represent a general strategy for the brain to address the cost-accuracy trade-off when solving hierarchical decision-making problems. This also may have important implications for learning and belief updating in biological as well as artificial neural networks, in particular for networks that are aimed at learning a generative model (*e.g.,* deep belief networks).

The fact that subjects condition their estimate on their preceding decision does not imply that they are necessarily fully confident in their decision; we propose that they simply do so in order to remain self-consistent. Our results show that conditioning is statistically independent of the difficulty and thus on subjects' confidence in their discrimination judgment (i.e., their psychometric function). However, it remains an interesting open question particularly in context of the ongoing discussion about decision confidence (*Kepecs et al., 2008*; *van den Berg et al., 2016*; *Fleming and Daw, 2017*) whether or not conditioning improves subjects' confidence in their subsequent estimate since it leads to a reduced posterior distribution.

Another interesting question is whether an explicit categorical commitment is necessary to induce self-consistent conditioning or whether the brain always, and thus implicitly, performs conditioned inference (*Ding et al., 2017*). This question is difficult to answer because without explicit access to a subject's trial-by-trial categorical judgment, differences in the subsequent feature inference process are often hard to distinguish statistically. Only in special cases when, for example, a hard discrimination boundary is present, these differences have a clear behavioral signature that can be identified (e.g., the characteristic repulsive estimation pattern in Experiment 2 of *Zamboni et al., (2016)*). Identification is further complicated because repulsive biases may also have other causes such as the efficient adaptation of sensory encoding resources (*Wei and Stocker, 2015*), which likely takes place during perceptual learning (e.g., *Szpiro et al., (2014)*]; see also *Wei and Stocker, (2017)*]). In fact, it may well be that implicit self-consistent inference is the fundamental process by which the brain solves inference problems, yet its behavioral characteristics are simply not often apparent. For example, we expect self-consistent conditioning to implicitly occur in object recognition: when an observer recognizes an object as an 'apple' the percept of the object's features (e.g., the color) will

automatically be conditioned on that recognized category. In order to detect the effects of this conditioning in perceptual behavior, however, we would need to know the specifics of the learned generative models over the object categories, which is typically a difficult problem. In other situations, such as in a typical psychophysical experiment with its sparse and artificial stimuli and little context, the observer may simply be given little incentive to interpret the stimulus within a hierarchical representation (generative model). Because the self-consistent inference model over a flat generative model is identical to an optimal Bayesian observer model, the large number of studies that have shown that perception is well explained as optimal Bayesian inference may actually not be conclusive; their data is equally well explained by the self-consistent inference model! This is obviously a strong hypothesis that needs further experimental evaluation.

Our results show interesting parallels to many well-known bias phenomena in cognition and economics, such as confirmation bias (*Nickerson, 1998*), biases associated with cognitive consistency (*Brehm, 1956*; *Abelson, 1968*) and dissonance (*Festinger, 1957*; *Festinger and Carlsmith, 1959*; *Egan et al., 2010*; *Sharot et al., 2010*), as well as loss aversion and the sunk cost fallacy (*Kahneman and Tversky, 1984*). Our findings seem also aligned with results from human probability judgments over hierarchical representations which found that subjects rather follow individual probability branches than to resolve the entire probability tree (*Lagnado and Shanks, 2003*). It will be interesting to explore to what degree the proposed self-consistent model generalizes to these cognitive phenomena and is able to provide a parsimonious, quantitative explanation.

Finally, results from recent physiological recordings in primates suggest not only that decision-making is associated with rapid cortical state-changes (*Meindertsma et al., 2017*) but also that decision-related signals are fed back along the perceptual processing pathway all the way to early sensory areas (*Nienborg and Cumming, 2009*; *Siegel et al., 2015*). The proposed self-consistent observer model provides a novel computational interpretation of these neural signals: at the moment a decision is reached the belief state of the brain rapidly changes (in favor of the choice made (*Peters et al., 2017*) and is fed back to ensure that the perceptual inference process remains consistent across the different cortical levels of representation at any moment in time. The self-consistent model may prove a useful hypothesis to constructively explore the function and purpose of such decision-related signal flows in the brain. Future work needs to explore how exactly our model formulation can be interpreted at a more mechanistic neural level (*Luu and Stocker, 2016a*). Various theoretical frameworks have been proposed for how the brain might perform Bayesian inference (e. g., *Ma et al., (2006)*; *Simoncelli, (2009)*; *Wei and Stocker, 2012*; *Pitkow and Angelaki, (2017)*). It remains an interesting challenge to investigate how these frameworks can incorporate the self-consistency constraint that we propose here, in particular the process of quickly and flexibly imposing a conditional prior.

## Materials and methods

### Experimental setup

Ten subjects with normal or corrected-to-normal vision (six males, four females; one non-naïve) participated in the experiments. One subject (male) was excluded from the analysis because he failed to correctly execute the estimation task. All subjects provided informed consent. The experiments were approved by the Institutional Review Board of the University of Pennsylvania under protocol #819634.

### General methods

Subjects sat in a dimmed room in front of a special purpose computer monitor (VIEWPixx3D, refresh rate of 120 Hz and resolution of 1920 x 1080 pixels). Viewing distance was 83.5 cm and enforced with a chin rest. We programmed all experiments in Matlab (Mathworks, Inc.) using the MGL toolbox (http://justingardner.net/mgl) for stimulus generation and presentation. The code was run on an Apple Mac Pro computer with Quad-Core Intel Xeon 2.93 GHz, 8 GB RAM. Subjects were asked to fixate a fixation dot whenever it appeared on the screen. Before subjects did the main experiments, they had 2–3 training sessions during which they familiarized themselves with the discrimination and the estimation task. After that, every subject either completed 1800 trials in 3–4 sessions for Experiment 1 or completed 3600 trials in 6–8 sessions for Experiment 2 and 3. This is equivalent to 40 trials

per every one of the 15 stimulus orientations and the three noise conditions. Sessions lasted approximately 50 min. Subjects used a gamepad (Sony PS4 Dualshock) as input device: they reported their decision in the discrimination task by pressing the corresponding trigger button (left for 'ccw', right for 'cw'), and used the analog joystick of the gamepad to indicate their perceived stimulus orientation by adjusting a reference line and then confirming their estimate with an additional button press. Screen background luminance was 40 cd/$\mathrm{m}^2$ and mean stimulus luminance was 49 cd/$\mathrm{m}^2$.

## Experiment 1

Five subjects (S1-5) participated in Experiment 1. In each trial, subjects viewed a white fixation dot (diameter: $0.3^o$) and two black marks (length: $3^o$, distance from fixation: $3.5^o$) indicating a discrimination boundary whose orientation was randomly chosen in each trial. After 1300 ms, the orientation stimulus consisting of an array of white line segments (length: $0.6^o$) was presented for 500 ms. The array consisted of two concentric circles of lines centered on the fixation: the outer (diameter: $3.8^o$) contained 16 line segments and the inner (diameter: $1.8^o$) contained eight line segments. Small random variations (from $-0.15^o$ to $0.15^o$) were independently added to the x-y coordinates of each line segment. The orientation of every line segment was drawn from a Gaussian distribution with mean given as one of 15 stimulus orientations relative to the boundary (from $-21^o$ to $21^o$ in steps of $3^o$) and standard deviation $\sigma$ as one of 3 values ($0^o, 6^o$ and $18^o$). After the stimulus disappeared, subjects were asked to indicate whether the overall orientation of the array was clockwise or counter-clockwise of the boundary. If subjects responded within 4 s, they then were also asked to indicate their perceived stimulus orientation. Otherwise the current trial was skipped and added to the back of the trial list. Every trial was followed by a randomly chosen inter-trial interval of 300-600 ms duration (blank screen; mean background luminance).

## Experiment 2

Five subjects (S1 and S6-9) participated in Experiment 2. The procedure was identical to Experiment 1 except that at the beginning of every trial, subjects were reminded of the stimulus distribution by presenting a prior cue consisting of a gray arc for 800 ms. The arc (width: $0.2^o$, eccentricity from fixation: $3.5^o$) spanned the range $\pm21^o$ relative to the discrimination boundary indicating the total true stimulus distribution. Subjects were instructed that stimulus orientation was guaranteed to occur anywhere within this range with equal probability.

## Experiment 3

The same five subjects that participated in Experiment 2 also participated in Experiment 3. The procedure was identical to Experiment 2 except for the following: First, the prior cue was present only for 500 ms, after which it was reduced to a colored arc that only spanned the orientation range at the side of the discrimination boundary where the stimulus orientation in this trial would occur. This colored cue indicated the correct answer ('cw' or'ccw') and was shown for 300 ms. Its color (red or green) was randomly assigned and uncorrelated with the stimulus orientation. Second, instead of the orientation discrimination task, subjects were tasked to recall the color of the cue.

## Control Experiment (Motor noise)

As part of the training, all subjects participated in a control experiment that allowed us to estimate subjects' individual motor noise levels. They were first presented with a fixation dot and the boundary (like in Experiment 1–3). After that, they were presented with an orientation stimulus consisting of a single white line (like the reference line in Experiment 1–3, see for example *Figure 1a*) for 500 ms. They then had to reproduce the stimulus by adjusting said reference line with the analog joystick of the gamepad. After subjects pressed a button to confirm their response, they received feedback in form of a green reference line indicating the true stimulus orientation. This control experiment consisted of 600 trials. The boundary orientation was uniformly sampled around the-circle. The stimulus orientation was uniformly distributed around the boundary as in Experiment 1–3. We used the measured standard deviation $\sigma_0$ in subjects' estimates as a direct measure of subjects' individual motor noise levels. *Figure 3—figure supplement 1* shows the measured $\sigma_0$ for all nine subjects. We used these measured levels as fixed parameters in all our model fits and predictions,

assuming that motor noise is Gaussian and is uniform and independent of the reproduced orientation relative to the discrimination boundary.

## Self-consistent Bayesian observer model

The model is formulated as an observer that performs Bayesian inference over the hierarchical generative model shown in *Figure 3a*. The observer is assumed to solve the two perceptual inference tasks in sequence: the observer first judges whether the stimulus orientation was clockwise or counter-clockwise of a random discrimination boundary, and then performs an estimate of the actual stimulus orientation. The key feature of the model is that the inference process associated with the estimate is *conditioned* on the observer's preceding categorical judgment. As such, the observer treats their own, subjective categorical judgment as if it was a correct statement about the stimulus (see Experiment 3). In the following we describe the Bayesian formalism of this conditioned task sequence.

### Discrimination judgment

Let $\theta$ be the true stimulus orientation relative to the discrimination boundary, $m$ the noisy sensory measurement of the stimulus orientation at presentation, and $C \in \{'cw','ccw'\}$ the categorical decision variable indicating whether the stimulus orientation is clockwise (cw) or counter-clockwise (ccw) of the discrimination boundary. Assuming a symmetric loss structure (i.e., no benefit for one correct decision over the other), the observer solves the categorical decision task by picking the category with maximal posterior probability given the sensory measurement $m$, thus

$$\hat{C}(m) = \mathrm{argmax}_{C \in \{'cw','ccw'\}} p(C|m). \tag{1}$$

The decision process $\hat{C}(m)$ is a deterministic mapping from any particular $m$ to either one of the two categories. The posterior distribution $p(C|m)$ is given as

$$p(C|m) \propto p(m|C)p(C), \tag{2}$$

where $p(C)$ is the prior probability and $p(m|C)$ the likelihood over the choices. We can obtain this likelihood by marginalizing the stimulus likelihood over all stimulus orientations, that is

$$p(m|C) = \int p(m|\theta)p(\theta|C)d\theta. \tag{3}$$

The stimulus likelihood is fully determined by the noise characteristics of the sensory measurement $m$, thus by the distribution $p(m|\theta)$ of sensory measurements $m$ for any given stimulus orientation $\theta$. Finally, the model's prediction of the psychometric function in the decision task (e.g., 'Fraction cw' - see *Figure 2b*) is obtained by marginalizing the mapping (*Equation 1*) for a particular choice over the sensory measurement distribution,

$$p(\hat{C} = 'cw'|\theta) = \int p(\hat{C} = 'cw'|m)p(m|\theta)dm. \tag{4}$$

### Estimation task

Following the categorical judgment, the observer then solves the estimation task by computing the mean of the posterior distribution (i.e., assuming a loss function that minimizes squared error) over $\theta$. In contrast to the independent observer, however, we assume that the posterior probability is *conditioned on the observer's own choice $\hat{C}$* in the preceding categorical decision task. Because the stimulus has long disappeared by the time the subject performs the estimation task (see experimental design), we formulate the posterior on a memorized version of the sensory measurement. With $m_m$ referring to a noisy recall sample from working memory of the measurement $m$ (doubly stochastic) we write the optimal conditioned estimate as

$$\hat{\theta}(m_m, \hat{C}) = \int \theta\, p(\theta|m_m, \hat{C})d\theta \tag{5}$$

with the posterior

$$p(\theta|m_m, \hat{C}) = \frac{p(m_m|\theta, \hat{C})p(\theta|\hat{C})}{p(m_m|\hat{C})}. \tag{6}$$

The estimate $\hat{\theta}(m_m, \hat{C})$, even though the result of Bayesian inference, describes a deterministic mapping from any particular $m_m$ to an estimate. However, we obtain two different formulations for the estimate, one for each potential categorical judgment.

With *Equation 5* the distribution of the estimates can be computed based on the distribution of the memorized sensory measurement

$$p(m_m|\theta, \hat{C}) = \int p(m_m|m)p(m|\theta, \hat{C}(m))dm. \tag{7}$$

Note that the above marginalization is limited to measurements $m$ that led to the particular categorical judgment $\hat{C}(m)$ (as given by *Equation 1*).

The model's description of the conditioned distribution of the estimates $p(\hat{\theta}|\theta, \hat{C})$ is obtained by a variable transformation for the conditional measurement distribution $p(m_m|\theta, \hat{C})$, that is substituting $m_m$ with the estimate $\hat{\theta}(m_m, \hat{C})$ (*Equation 5*). Finally, the model's prediction of the entire distribution of the estimates $p(\hat{\theta}|\theta)$ (i.e., the density plots shown in, e.g., *Figure 2c*) is given by marginalizing over the decision outcomes, thus

$$p(\hat{\theta}|\theta) = \sum_{\hat{C}} p(\hat{\theta}|\theta, \hat{C})p(\hat{C}|\theta). \tag{8}$$

## Estimation task with known correct category assignment (Experiment 3)

If the observer knows the category assignment upfront (as in Experiment 3), the above formulation of solving the estimation task slightly changes in that *Equations 5–8* are conditioned on the correct category assignment $C$ rather than a the inferred category $\hat{C}$. In particular, this changes the marginalization over $m$ in *Equation 7* such that it is no longer limited to values of $m$ that are consistent with a desired category assignment (according to $\hat{C}(m)$), and also *Equation 8* where the sum is over the actual category probability $p(C|\theta)$ rather than the inferred probability $p(\hat{C}|\theta)$. As a result, the predicted biases for identical model parameters are slightly smaller when the observer knows the correct category assignment compared to when the category has to be inferred first.

## Specific assumptions defining the generative model

We made the following specific assumptions in defining the components of the generative model (*Figure 3a*):

- We set the category prior $p(C) = 0.5$ because the two choices are equally likely in all our experiments.
- The categorical stimulus prior $p(\theta|C)$ was assumed to reflect subjects' individual expectations about the experimental distribution of stimulus orientations. We modeled this prior to be identical but mirrored around the discrimination boundary for the two choices. More specifically, we assumed it to be uniform over the angle $\alpha$ relative to the boundary with a smooth cosine fall-off from the uniform density value to zero over the additional angle $\beta$. We then defined the prior width $w_p$ (see *Figures 4b* and *7c*) as the total angle relative to the boundary where the prior density decreased to half of its uniform value, that is $w_p = (\alpha + 2/3\beta)$.
- We assumed the sensory measurements $m$ to reflect noisy samples of the true stimulus orientation $\theta$, with $p(m|\theta)$ to be a Gaussian with mean $\theta$ and standard deviation $\sigma_s$ that is monotonically depending on the array distribution width $\sigma$ of the orientation stimulus. Although $\sigma_s$ was assumed to be subject dependent and thus a free parameter, we assumed that across experiments $\sigma_s$ was the same for a given subject and stimulus noise condition.
- We treat the sensory evidence in the estimation task $m_m$ as being a sample from a Gaussian with mean $m$ (original sensory measurement) and standard deviation $\sigma_m$. We assumed that $\sigma_m$ is independent of stimulus uncertainty yet is different for different subjects.

## Independent Bayesian observer model

The 'independent' observer model, as defined in our paper, is formulated on the same generative model as the self-consistent observer model (*Figure 3a*) and thus has identical model parameters. The only formal differences are that

- the posterior over stimulus orientation is not conditioned on the discrimination judgment $\hat{C}$ (as in *Equation 6*), thus

$$p(\theta|m_m) = \frac{p(m_m|\theta)p(\theta)}{p(m_m)} \quad \text{with} \quad p(\theta) = \sum_C p(\theta|C)p(C),\tag{9}$$

- marginalization over the memorized sensory signal is not limited to measurements that are in agreement with a particular discrimination judgment $\hat{C}$ (as in *Equation 7*), thus

$$p(m_m|\theta) = \int p(m_m|m)p(m|\theta)dm.\tag{10}$$

Having the same parameters as the self-consistent observer model allows a direct log-likelihood comparison in judging the goodness-of-fit.

## Model fits

We jointly fit the model to the data of both the decision and estimation task by maximizing the likelihood of the model given the data:

$$p(D|\rho) = \prod_{i=1}^{n} P(D_i|\rho) = \prod_{i=1}^{n} P(\hat{C}_i|\rho,\theta)p\left(\hat{\theta}_i|\hat{C}_i,\rho,\theta\right)\tag{11}$$

where $D$ is the data, $\rho$ represents the parameters of the model, $\theta$ is the true orientation, $\hat{C}_i$ is the decision outcome, $\hat{\theta}_i$ is the orientation estimate, $i$ is the trial index and $n$ is the number of trials.

For all fits, we only excluded trials with inconsistent response pairs (i.e., trials in which subjects' discrimination judgment and estimate were not consistent in terms of their categorical identity; approximately 4% of the trials.). As we demonstrate (*Figure 8*), the fractions and the bias characteristics of these inconsistent trials can be fully predicted based on the fit model parameters to the consistent data, assuming that they are caused by motor noise and lapses (*Figure 8*).

Subjects' motor noise was accounted for by assuming that the recorded orientation estimates follow the distributions of estimates $p(\hat{\theta}|\theta)$ (*Equation 8*) convolved with the motor noise kernel. We assumed motor noise to be additive Gaussian with a standard deviation $\sigma_0$ that was individually determined for each subject from the control experiment (see above). Motor noise levels across subjects were fairly similar with an average $\sigma_0 = 4.3$ degrees. *Figure 3—figure supplement 1* shows measured noise levels for all subjects.

Our model fit contained a total of 6 free parameters:

- standard deviations $\sigma_s$ for the three noise levels of the stimuli (additive Gaussian noise).
- standard deviation $\sigma_m$ for memory noise (additive Gaussian).
- two parameters $\alpha, \beta$ for the prior distribution over stimulus orientation, defining its uniform range and smoothness, respectively.

The Nelder-Mead simplex algorithm was used to minimize the term $-log(p(D|\rho))$. Twenty iterations of the optimization procedure were performed using randomized initial parameter values in order to obtain the best fitting model.

## Model fit to data by *Zamboni et al., (2016)*

*Experiment 1:*

For the condition where the decision boundary was always present, we fit the data with exactly the same model assumptions as we used for fitting the data from our Experiment 1. We added motor noise as a free parameter because *Zamboni et al., (2016)* did not use a control experiment to determine the motor noise. For the condition where the decision boundary was removed after subjects

did the discrimination task, we assumed that the observer had to rely on a noisy memory recall of the true boundary orientation $\theta_b$ when performing the estimation task. As a result, the conditioned prior varies in every trial depending on that memory recall. We assumed the recalled orientation to be Gaussian distributed around the true boundary orientation with a standard deviation $\sigma_b$ that was a free parameter. Because the same group of subjects run both versions of the experiments, we first fit the self-consistent model to the data from the boundary-present condition and then used those parameters to fit the data in the boundary-absent condition with the only free parameter being $\sigma_b$ (*Figure 10—figure supplement 1*). When computing the goodness-of-fit (*Figure 10—figure supplement 2*) we assumed the independent observer model to have the same additional noise parameter $\sigma_b$.

### Experiment 2

Although the discrimination boundary was present throughout the entire trial, subjects were only asked to perform the estimation task. Furthermore, unknown to the subjects, the decision boundary was either kept the same or shifted six degrees (cw or ccw). For our model fit, we assumed that subjects, implicitly performed the discrimination task and then subsequently conditioned the estimation process on that implicit decision as described by the proposed self-consistent observer model. Thus we fit the data with exactly the same model assumptions used to fit the data from our Experiment 1, with the addition that for the conditions where the decision boundary was shifted, the conditioned prior was shifted accordingly. The fits shown in *Figure 10*, *Figure 10—figure supplement 3* are joint fits to data from all three shifted conditions.

## Code availability

Computer code (MATLAB) providing model simulations for all three experiments is freely available at https://github.com/cpc-lab-stocker/Self-consistent-model (*Luu and Stocker, 2018*). A copy is archived at https://github.com/elifesciences-publications/Self-consistent-model.

## Acknowledgements

This work was supported by the National Science Foundation of the United States of America (CAREER award 1350786). A preliminary manuscript describing some of the results has been posted on an electronic archive (*Luu and Stocker, 2016b*). We thank Elisa Zamboni and colleagues for providing their experimental data. We are grateful to the members of the computational perception and cognition laboratory for many fruitful discussions about self-consistent inference.

## Additional information

### Funding

| Funder | Grant reference number | Author |
| --- | --- | --- |
| National Science Foundation | NSF CAREER award 1350786 | Alan A Stocker |

The funders had no role in study design, data collection and interpretation, or the decision to submit the work for publication.

### Author contributions

Long Luu, Conceptualization, Data curation, Software, Formal analysis, Validation, Investigation, Visualization, Methodology, Writing—review and editing; Alan A Stocker, Conceptualization, Resources, Software, Formal analysis, Supervision, Funding acquisition, Validation, Investigation, Visualization, Methodology, Writing—original draft, Project administration, Writing—review and editing

### Author ORCIDs

Long Luu http://orcid.org/0000-0003-1761-3105
Alan A Stocker http://orcid.org/0000-0002-2041-1515

### Ethics

Human subjects: All subjects provided informed consent. Experiments were approved by the Institutional Review Board of the University of Pennsylvania under protocol #819634.

### Decision letter and Author response

Decision letter https://doi.org/10.7554/eLife.33334.029
Author response https://doi.org/10.7554/eLife.33334.030

## Additional files

### Supplementary files

• Source data 1. Data for Experiment 1.
DOI: https://doi.org/10.7554/eLife.33334.022

• Source data 2. Data for Experiment 2.
DOI: https://doi.org/10.7554/eLife.33334.023

• Source data 3. Data for Experiment 3.
DOI: https://doi.org/10.7554/eLife.33334.024

• Transparent reporting form
DOI: https://doi.org/10.7554/eLife.33334.025

### Data availability

Data (human psychophysics) is provided as Source data files 1-3.

The following previously published dataset was used:

| Author(s) | Year | Dataset title | Dataset URL | Database, license, and accessibility information |
|---|---|---|---|---|
| Zamboni E, Ledgeway T, McGraw PV, Schluppeck D | 2016 | Data from: Do perceptual biases emerge early or late in visual processing? Decision-biases in motion perception. | http://dx.doi.org/10.5061/dryad.ms84h | Available at Dryad Digital Repository under a CC0 Public Domain Dedication |

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
