## [Decision Letter]

Thank you for submitting your article "Post-decision biases reveal a self-consistency constraint in perceptual inference" for consideration by *eLife*. Your article has been favorably evaluated by Timothy Behrens (Senior Editor) and three reviewers, one of whom, Tobias H Donner (Reviewer #1), is a member of our Board of Reviewing Editors. The following individuals involved in review of your submission have agreed to reveal their identity: Mehrdad Jazayeri (Reviewer #2); Bruce G Cumming (Reviewer #3); Valentin Wyart (Reviewer #4).

The reviewers have discussed the reviews with one another and the Reviewing Editor has drafted this decision to help you prepare a revised submission.

Summary:

Your paper reproduces influential previous findings on how combining a discrimination task with an estimation task produces systematic biases in estimation (Jazayeri and Movshon, 2007). The previous study had explained the estimation bias as the result of an optimal readout of sensory evidence for the discrimination task (henceforth referred to as 'readout model'). You provide a different explanation in terms of a self-consistent Bayesian observer. You perform two additional experiments (cueing the stimulus range, and providing the answer to a discrimination task), and show that the results follow the predictions of your model well.

All reviewers thought that your paper makes a compelling case that your self-consistent Bayesian observer provides an excellent account of all the data, including variation between individual observers. The reviewers also appreciated your discussion of the potential adaptive function of this self-consistency bias and its relation to 'cognitive dissonance' phenomena described in the social psychology literature.

One issue contentious among reviewers was whether the conceptual advance provided by your paper is sufficient to warrant publication in *eLife*. The reviewers converged on the conclusion that (with the revisions requested below) the paper would be borderline for *eLife*, but that the case could be made for it being above threshold. Another issue was whether or not it was necessary that you report a formal head-to-head comparison between your self-consistent observer and the readout model by Jazayeri and Movshon. The reviewers eventually agreed that it would be unreasonable to ask you for truly defensible head-to-head comparison. Doing so would require that you develop a complete algorithmic model, which would demand making many assumptions about how a prior is encoded and how it would interact with readout weights. It would also require you to make another set of assumptions about how factors such as prior knowledge ought to be included in the old readout model, which would weaken your argument and blur the distinction between the two models.

Essential revisions:

1) The paper is written in a way that makes it difficult for the reader to appreciate which part is replication and which part is truly novel. Please revise the Abstract and Introduction to clarify this, focusing on the following points.

The original observation of self-consistent behavior was made several decades ago. Jazayeri and Movshon reported this phenomenon in perception, and since then, several studies have replicated the finding (e.g., Szpiro et al., 2014; Zamboni et al., 2017). Stocker and Simoncelli developed and published the self-consistent observer model in NIPS in 2008. The current model is almost an exact replica of that previously published one. Experiment 1 is a replica of the original Jazayeri and Movshon experiment, with the exception of using orientation rather than motion direction. The main novel contribution of your present paper are the empirical tests (Experiments 2 and 3).

2) Please add a discussion of whether the original readout model (Jazayeri and Movshon, 2007) cannot account for your data. The reviewers felt that changes in the prior on orientation width (caused by cuing) might produce similar effects in their model, without having to invoke self-consistency; or that providing the observer with veridical information about the stimulus category might naturally change the percept in the traditional Bayesian way. The reviewers thought it would be important to make clear to what extent we now have two equally good explanations of the phenomenon. Specifically, the reviewers would like you to rework the paper to clarify the following three things in the Discussion:

- Explain that the two models are tackling the question at different levels. This would help the paper highlight an important conceptual point that the previous work failed to convey.

- Explain that Experiments 2 and 3 may not be decisive for arbitrating between the two models. Even without this, the two experiments highlight an important and complementary point that when other sources of information are available (prior information and/or other cues), these sources could influence the decoding strategy (in both models). This is particularly relevant because if one subsumes the effects of prior into the readout weights, then the difference in the two models become a matter of semantics and would diminish the intuitive interpretation that makes the paper appealing.

- Explain that to properly compare the two models, future research has to (1) develop an algorithmic version of their model, and (2) consider how prior information could be worked into the old readout model. This would allow the paper to inform the path forward for meaningful head-to-head comparisons. In this context, it would be useful to discuss what it means to suddenly abolish a portion of the posterior. This can be done conveniently in a model but how does it fit with what we know about how these computations are carried out by the brain?

3) Two recent studies investigated the same estimation bias and arrived at conclusions that are different from yours. Those should be addressed in more detail. Specifically:

- Zamboni et al. (2017) established an important role of the presence and orientation of the reference in inducing estimation biases. The reference can induce the bias even without any categorical choice. This seems to at odds with your claim that choices *cause* the estimation bias. You briefly touch on this study in Discussion, claiming that the current self-consistent observer model can account for this. Given that this issue is critical for your conclusion, you should fit your model to the Zamboni et al. data (which are available).

- Results from Szpiro et al. (2014) suggest that the phenomenon is learned through practice. This suggests that it cannot be straightforwardly attributed to a conditionalization. Additionally, in the same study, it was found that the bias manifests itself differently in the perceptual and motor domains. If we have to come up with a different model for each domain, then the exercise of using a Bayesian approach seem less revealing. It is disappointing that this study is not cited given that the empirical observations are directly against the proposed hypothesis.

4) A key component of your model is conditionalization of the posterior according to prior choice. You motivate this by noting that this allows subjects to be self-consistent. However, in most realistic conditions, the behavior would remain consistent even without this component – the conditionalization just moves the estimate away from the boundary, it does change sides. Since the model doesn't actually increase self-consistency, please describe more carefully what motivates it.

5) Please provide quantitative measures about the quality of fits. Your model fits look compelling, but the assessment is only qualitative at present.

---

## [Author Response]

Essential revisions:1) The paper is written in a way that makes it difficult for the reader to appreciate which part is replication and which part is truly novel. Please revise the Abstract and Introduction to clarify this, focusing on the following points.The original observation of self-consistent behavior was made several decades ago. Jazayeri and Movshon reported this phenomenon in perception, and since then, several studies have replicated the finding (e.g., Szpiro et al., 2014; Zamboni et al., 2017). Stocker and Simoncelli developed and published the self-consistent observer model in NIPS in 2008. The current model is almost an exact replica of that previously published one. Experiment 1 is a replica of the original Jazayeri and Movshon experiment, with the exception of using orientation rather than motion direction. The main novel contribution of your present paper are the empirical tests (Experiments 2 and 3).

We have revised Abstract and Introduction in order to clarify what the novel contributions of the paper are. We now clearly express that the main contribution of the paper is to test the self-consistent observer hypothesis (Stocker and Simoncelli, 2007) by formulating an extended model and validating it with data from new psychophysical experiments (Experiment 1: Replica of Jazayeri and Movshon (2007) but with different perceptual variable) and existing data (Zamboni et al., 2016, new).

Note that our original model formulation was a theoretical proposal (Stocker and Simoncelli, 2007, NIPS conference proceedings); it has never been directly tested against data. It lacked the assumption that the inference process involves working memory which, as our results show, is not only important for an accurate account of the data but also provides a more meaningful motivation for why self-consistent inference is behaviorally beneficial (see new Figure 9).

2) Please add a discussion of whether the original readout model (Jazayeri and Movshon, 2007) cannot account for your data. The reviewers felt that changes in the prior on orientation width (caused by cuing) might produce similar effects in their model, without having to invoke self-consistency; or that providing the observer with veridical information about the stimulus category might naturally change the percept in the traditional Bayesian way. The reviewers thought it would be important to make clear to what extent we now have two equally good explanations of the phenomenon. Specifically, the reviewers would like you to rework the paper to clarify the following three things in the Discussion:- Explain that the two models are tackling the question at different levels. This would help the paper highlight an important conceptual point that the previous work failed to convey.- Explain that Experiments 2 and 3 may not be decisive for arbitrating between the two models. Even without this, the two experiments highlight an important and complementary point that when other sources of information are available (prior information and/or other cues), these sources could influence the decoding strategy (in both models). This is particularly relevant because if one subsumes the effects of prior into the readout weights, then the difference in the two models become a matter of semantics and would diminish the intuitive interpretation that makes the paper appealing.- Explain that to properly compare the two models, future research has to (1) develop an algorithmic version of their model, and (2) consider how prior information could be worked into the old readout model. This would allow the paper to inform the path forward for meaningful head-to-head comparisons. In this context, it would be useful to discuss what it means to suddenly abolish a portion of the posterior. This can be done conveniently in a model but how does it fit with what we know about how these computations are carried out by the brain?

We have substantially extended and revised our discussion of the relationship between our self-consistent inference model and the “read-out” model by Jazayeri and Movshon (2007). We addressed the points outlined above as follows (subsection “Alternative interpretations”):

- We now explain that a direct model comparison is difficult because the models are expressed at different levels (computational vs. mechanistic/neural). We also clarify that beyond this difference, the two models are fundamentally different in how they consider the interaction between the discrimination and the estimation task: our model assumes that the discrimination judgment causally affects the estimation process, while the read-out model assumes a feed-forward process where the dependency is established only via the shared read-out profile.

- We now explain that the results of Experiment 2 and 3 may be compatible with a readout interpretation, although that would require relaxing the original rationale for the nonuniform read-out profile (optimized for discrimination task). Future work is necessary to derive principled ways of how stimulus prior information (as presented by cues in Experiment 2 and 3) is incorporated into the weighting profile.

- We now discuss how the impact of working memory noise on trial consistency (new Figure 9) provides further evidence in favor of our model.

- We now also stress that a validation of the read-out model, or any alternative model for that purpose, would ultimately need to take full advantage of the richness in the obtained data from such task sequences, i.e., a valid model should be able to accurately explain the full distributions of the trial response-pairs (consisting of both, the category choice and the estimate).

- Finally, we briefly discuss the challenges in deriving a more mechanistic, neural implementation of our model in future work (subsection “General implications for computational and neural models of decision-making”, last paragraph). Note, that our recent experimental results suggest that conditioning on the category choice is better thought of as imposing a categorical prior rather than abolishing sensory (posterior) information (Luu and Stocker, 2016).

3) Two recent studies investigated the same estimation bias and arrived at conclusions that are different from yours. Those should be addressed in more detail. Specifically:- Zamboni et al. (2017) established an important role of the presence and orientation of the reference in inducing estimation biases. The reference can induce the bias even without any categorical choice. This seems to at odds with your claim that choices cause the estimation bias. You briefly touch on this study in Discussion, claiming that the current self-consistent observer model can account for this. Given that this issue is critical for your conclusion, you should fit your model to the Zamboni et al. data (which are available).- Results from Szpiro et al. (2014) suggest that the phenomenon is learned through practice. This suggests that it cannot be straightforwardly attributed to a conditionalization. Additionally, in the same study, it was found that the bias manifests itself differently in the perceptual and motor domains. If we have to come up with a different model for each domain, then the exercise of using a Bayesian approach seem less revealing. It is disappointing that this study is not cited given that the empirical observations are directly against the proposed hypothesis.

We thank the reviewers for this suggestion. We added a new section (including new Figure 10) showing how our model can also account for the experimental data by Zamboni et al. (2016). In particular, we show that the data is consistent with self-consistent inference if we assume subjects made implicit categorical judgments at the mere presence of the discrimination boundary (Experiment 2). The assumption seems reasonable given that subjects first participated in Experiment 1 (where they had to explicitly make a decision) before participating in Experiment 2, and thus have likely internalized the discrimination process. It also matches our own experience in running these types of experiments. Note also that we further discuss the general possibility of implicit self-consistent conditioning in the Discussion (subsection “General implications for computational and neural models of decision-making”, fourth paragraph).

The experiments by Szpiro et al. (2014) may or may not involve the type of conditioning we propose. This is difficult to determine given that subjects were neither tasked to perform a discrimination task nor shown an explicit discrimination boundary. As we discuss, it is difficult to recognize perceptual signatures (e.g., bimodal patterns) in the data under such conditions. However, it is important to realize that self-consistent conditioning is not the only cause for repulsive biases; other effects may contribute as well. For example, we have recently proposed that efficient coding principles can explain repulsive biases within a Bayesian framework (Wei and Stocker, 2015). In a recent paper we specifically interpret the results by Szpiro et al. (2014) within this theoretical framework (Wei and Stocker, 2017). Thus, we do not consider the results by Szpiro et al. (2014) as conflicting with our self-consistent model but rather that they are due to an additional, efficient coding mechanism, which, for the benefit of simplicity, we have not included in the current paper. We address this point in the Discussion (subsection “General implications for computational and neural models of decision-making”, fourth paragraph).

Finally, a discussion about the apparent differences in perceptual versus motor (smooth pursuit) bias seems beyond the scope of our paper, in particular since the reported differences focus on the effect of perceptual learning on the magnitude of the bias and not on the induction of the bias in the first place. Note that for both domains biases were actually repulsive pre- and post-learning (see Figure 3A and B in Szpiro et al. (2014)).

4) A key component of your model is conditionalization of the posterior according to prior choice. You motivate this by noting that this allows subjects to be self-consistent. However, in most realistic conditions, the behavior would remain consistent even without this component – the conditionalization just moves the estimate away from the boundary, it does change sides. Since the model doesn't actually increase self-consistency, please describe more carefully what motivates it.

Thank you for asking us to clarify this point. With a new analysis we now demonstrate that without conditioning on the category choice, model behavior would exhibit a substantial fraction of inconsistent trials due to working memory noise (see new Figure 9). The fraction reflects those trials where the sensory signal *m* (basis of the category choice) and its working memory copy *m_m_*(basis for the estimate) happen to be on different sides of the discrimination boundary. A comparison with the actual fraction of inconsistent trials in our data shows that they are much smaller than these predictions (and fully explained by lapses and motor noise; Figure 8). This would provide direct experimental evidence that subjects’ estimates are indeed conditioned on their preceding category choice if the working memory noise levels we extracted with our model fits were realistic. Future work has to independently confirm whether that is true, although the fit levels are well in line with previous results (Bays et al., 2011, e.g.).

We now also incorporate the results of this analysis in our discussion about the motivation and benefits of self-consistent inference. As a result, we also toned down the notion that self-consistent inference is necessarily irrational.

5) Please provide quantitative measures about the quality of fits. Your model fits look compelling, but the assessment is only qualitative at present.

We now provide log-likelihood analyses of the model fits in Figure 4—figure supplement 1, Figure 7—figure supplement 1, and Figure 10—figure supplement 3, respectively.